# RePaViT: Scalable Vision Transformer Acceleration via Structural Reparameterization on Feedforward Network Layers

Xuwei Xu [1 2]   Yang Li [3]   Yudong Chen [1]   Jiajun Liu [3 1]   Sen Wang [1 2]

## Abstract

We reveal that feedforward network (FFN) layers, rather than attention layers, are the primary contributors to Vision Transformer (ViT) inference latency, with their impact signifying as model size increases. This finding highlights a critical opportunity for optimizing the efficiency of large-scale ViTs by focusing on FFN layers. In this work, we propose a novel channel idle mechanism that facilitates post-training structural reparameterization for efficient FFN layers during testing. Specifically, a set of feature channels remains idle and bypasses the nonlinear activation function in each FFN layer, thereby forming a linear pathway that enables structural reparameterization during inference. This mechanism results in a family of **RePa**rameterizable **Vi**sion **T**ransformers (RePaViTs), which achieve remarkable latency reductions with acceptable sacrifices (sometimes gains) in accuracy across various ViTs. The effectiveness of our method scale consistently with model sizes, demonstrating greater speed improvements and progressively narrowing accuracy gaps or even higher accuracies on larger models. In particular, RePaViT-Large and RePa-ViT-Huge enjoy **66.8%** and **68.7%** speed-ups with **+1.7%** and **+1.1%** higher top-1 accuracies under the same training strategy, respectively. RePaViT is the first to employ structural reparameterization on FFN layers to expedite ViTs to our best knowledge, and we believe that it represents an auspicious direction for efficient ViTs. Source code is available at https://github.com/Ackesnal/RePaViT.

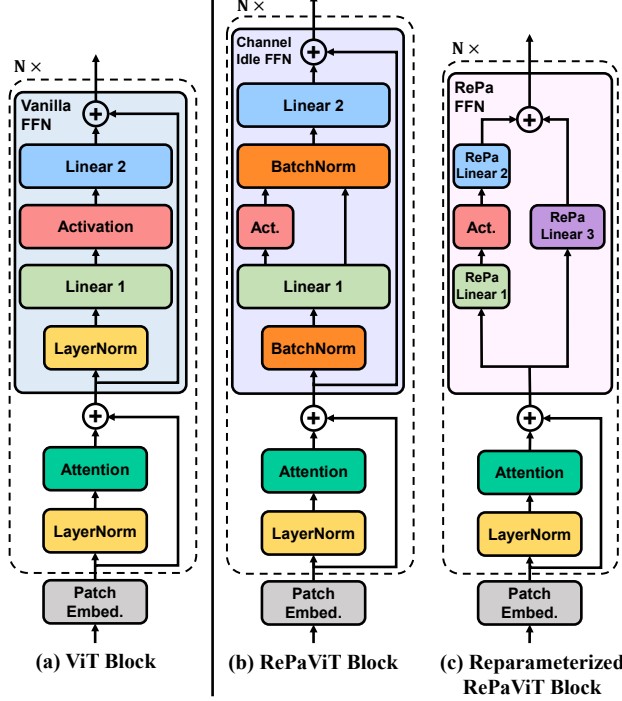

*Figure 1.* **RePaViT architecture.** (a) represents the vanilla ViT block. (b) illustrates our channel idle mechanism for FFN layers during training, where only a subset of channels are activated while the rest bridge a linear pathway. (c) shows the reparameterized RePaViT block during testing, where the number of parameters and computational complexity are significantly reduced.

[1]School of Electrical Engineering and Computer Science, The University of Queensland, Brisbane, Australia. [2]ARC Training Centre for Information Resilience (CIRES), The University of Queensland, Brisbane, Australia. [3]DATA61, CSIRO, Pullenvale, Brisbane, Australia.. Correspondence to: Jiajun Liu <ryan.liu@data61.csiro.au>, Sen Wang <sen.wang@uq.edu.au>.

*Proceedings of the $42^{nd}$ International Conference on Machine Learning*, Vancouver, Canada. PMLR 267, 2025. Copyright 2025 by the author(s).

## 1. Introduction

Vision Transformer (ViT) (Dosovitskiy et al., 2021) and its advanced variants (Touvron et al., 2021; Liu et al., 2021; Ryoo et al., 2021; Yu et al., 2022c; Liu et al., 2022; Dehghani et al., 2023) have achieved outstanding performance in various computer vision tasks. However, the high computational cost and memory demand of ViTs hinder their wide deployment in real-world scenarios, especially in computing resource-constrained environments.

To improve efficiency for ViTs, several techniques have been developed, such as token pruning (Rao et al., 2021; Liang et al., 2021; Kong et al., 2022a;b; Fayyaz et al., 2022) and

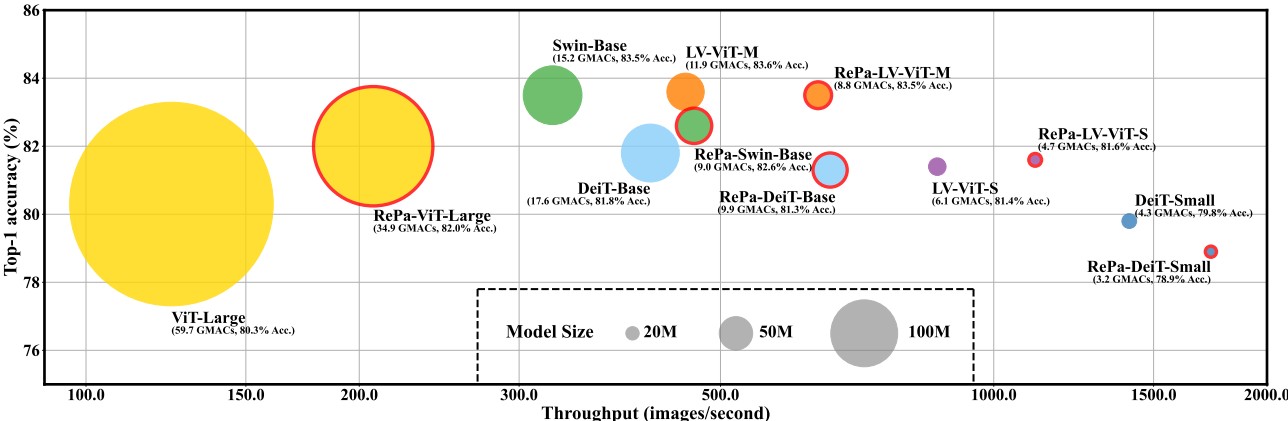

*Figure 2.* **Performance comparison of RePaViTs and their vanilla backbones.** RePaViTs (red circled) consistently achieve greater accelerations and smaller accuracy gaps when model sizes increase, showing the potential effectiveness in expediting large-scale ViTs. It is also worth noting that RePa-ViT-Large not only improves inference speed by more than 50% but also raises accuracy by 1.7%.

token merging (Bolya et al., 2023; Zong et al., 2022; Marin et al., 2023; Xu et al., 2024b; Kim et al., 2024) methods that gradually reduce the number of image tokens as the layer goes deep; hybrid architectures (Mehta & Rastegari, 2022a; Chen et al., 2022a; Maaz et al., 2022; Li et al., 2022; Zhang et al., 2023) that embed efficient convolutional neural networks (CNNs) into ViTs; and network pruning (Yu et al., 2022b;a; Yu & Xiang, 2023; Zhang et al., 2024; He & Zhou, 2024) methods that remove less important parameters while preserving performance. Meanwhile, knowledge distillation methods (Touvron et al., 2021; Hao et al., 2022; Wu et al., 2022; Chen et al., 2022b) are introduced to further optimize efficient ViTs' performance.

Despite growing interest in efficient ViTs, existing approaches often overlook structural reparameterization (Ding et al., 2019; 2021b; Zhu et al., 2023), a powerful network simplification technique widely used in CNNs. Structural reparameterization enables networks to adopt different structures during training and inference by merging multi-branch convolutions or adjacent BatchNorm (Ioffe & Szegedy, 2015) and convolution via linear algebra operations. This process allows a complex architecture during training to be compressed into a simpler structure for inference, thereby improving efficiency. Some recent research (Vasu et al., 2023a; Guo et al., 2024) has investigated structural reparameterization for ViTs by integrating elements from CNNs into ViTs and subsequently reparameterizing only these CNN components. However, little attention has been given to directly applying structural reparameterization to the intrinsic architecture of ViTs, particularly to their fundamental building blocks.

Among these building blocks, feedforward network (FFN) layers represent a promising yet underexplored target for applying structural reparameterization. A typical FFN layer consists of two consecutive linear projections with a nonlin-

ear activation function in between (*i.e.*, Figure 1(a)). The two linear projections can be potentially merged via structural reparameterization to reduce complexity during testing. Notably, reducing FFN complexity is particularly critical for improving the efficiency of ViTs. Despite their straightforward structure, FFN layers account for more than 60% of the total computational complexity in ViT models (Li et al., 2022; Mehta & Rastegari, 2022b). Furthermore, we observe that FFN layers contribute a substantial portion of the total latency in ViTs, with this contribution scaling up as the model size grows, as shown in Figure 3. These observations reflect the urgent demand for techniques to optimize FFN layers, especially for large-scale ViTs.

To facilitate structural reparameterization for FFN layers, in this work, we propose an innovative channel idle mechanism. Specifically, in each FFN layer, only a small subset of feature channels undergo the activation function to provide necessary nonlinearity while the rest channels remain idle, as shown in Figure 1(b). Consequently, these idle channels bridge a linear pathway through the activation function, enabling structural reparameterization during inference. Moreover, inspired by Yao et al. (2021), we substitute the LayerNorm (Lei Ba et al., 2016) with BatchNorm (Ioffe & Szegedy, 2015) and add another BatchNorm before the second linear projection. These BatchNorms can be reparameterized into their adjacent linear projection weights, which allows further reparameterization of the shortcut.

With the proposed channel idle mechanism, a family of **RePa**rameterizable **Vi**sion **T**ransformers (RePaViTs) are developed, whose FFN layers can be reparameterized to condensed structures during inference as Figure 1(c) shows. Extensive experiments on various ViTs have validated the effectiveness of our method, demonstrating its potential to enhance the applicablity of ViTs in resource-constrained environments. Moreover, as Figure 2 illustrates, the ex-

perimental results further indicate that our method delivers more significant acceleration and narrower performance disparity as the model complexity increases. In particular, RePaViT accelerates ViT-Large and ViT-Huge models by ~68% speed gain while even improving accuracy by 1~2% compared to their vanilla versions. This also demonstrates a transformative contribution, as many practical large-scale foundation models for computer vision tasks utilize ViTs as their backbones, such as CLIP (Radford et al., 2021; Cherti et al., 2023) and SAM (Kirillov et al., 2023). Moreover, our RePaViT achieves better trade-offs between speed improvement and accuracy compared to state-of-the-art network pruning methods.

To our best knowledge, RePaViT is the first method that successfully applies structural reparameterization on FFN layers for efficient ViTs, and achieves significant acceleration while having positive gains in accuracy instead of accuracy drops on large and huge ViTs with the same training strategies.

## 2. Related Work

### 2.1. Efficient Vision Transformer Methods

Vision Transformer (ViT) (Dosovitskiy et al., 2021) adapts the Transformer (Vaswani et al., 2017) architecture for computer vision, achieving success on various computer vision tasks. However, ViT suffers a substantial computational complexity. To alleviate the computational burden, several techniques that focus on structural design for efficient ViTs have been proposed. Spatial-wise token reduction methods are developed to identify less important tokens and subsequently prune (Rao et al., 2021; Liang et al., 2021; Kong et al., 2022a; Fayyaz et al., 2022; Xu et al., 2022; Meng et al., 2022; Tang et al., 2022; Xu et al., 2023) or merge (Bolya et al., 2023; Zong et al., 2022; Marin et al., 2023; Xu et al., 2024b; Kim et al., 2024) them during inference. As a result, the number of tokens participating in the self-attention computation is reduced. Meanwhile, hybrid architectures that combine self-attentions with computationally efficient convolutions (Graham et al., 2021; Mehta & Rastegari, 2022a; Chen et al., 2022a; Li et al., 2022; Cai et al., 2023; Vasu et al., 2023a; Zhang et al., 2023; Shaker et al., 2023) are introduced to reduce the computationally expensive self-attention operations while introducing regional biases into ViTs. In addition to hybrid ViTs, MetaFormer (Yu et al., 2022c) figures out that ViTs benefit from their architectural design, which consists of one token mixer layer and one multi-layer perception layer, and the token mixer can be replaced by more efficient operations, such as average pooling (Yu et al., 2022c) or linear projection (Tolstikhin et al., 2021). However, these approaches overlook the structural reparameterization method, which can effectively compress a network that contains consecutive linear transformations,

such as FFN layers in ViTs. Our work is the first to apply structural reparameterization on FFN layers for ViTs.

### 2.2. Structural Reparameterization

Structural reparameterization is an effective network simplification technique that is typically employed in multi-branch CNNs (Ding et al., 2019; Guo et al., 2020; Ding et al., 2021a;b). It converts an over-parameterized network block into a compressed structure during testing, thereby reducing the model complexity and increasing the speed for the inference stage. For instance, after reparameterizing its multi-branch convolutions and shortcuts into a single branch, RepVGG-B0 (Ding et al., 2021b) achieves 71% speed-up with no accuracy loss. Although some recent studies claim to adopt structural reparameterization for enhancing ViTs' efficiency (Vasu et al., 2023a; Wang et al., 2024; Tan et al., 2024), they primarily construct a hybrid architecture consisting of both convolutions and self-attentions and only perform reparameterization on the convolutional part. A recent state-of-the-art method, SLAB (Guo et al., 2024), proposes to progressively substitute LayerNorms in ViTs with BatchNorms and reparameterize BatchNorms into linear projection weights. Unlike these methods, we are the first to apply structural reparameterization on FFN layers.

## 3. Method

### 3.1. Latency Analysis

To understand the significance of improving efficiency for FFN layers, we profile the latencies of major components in several representative ViT models in Figure 3, including DeiT (Touvron et al., 2021), Swin Transformer (Liu et al., 2021) and ViT (Dosovitskiy et al., 2021). Figure 3 illustrates that FFN layers constitute a substantial portion of the total processing time, which escalates quickly as the model size increases. For instance, in the DeiT-Small model, FFN layers contribute to approximately 32.8% of the inference time, while in the DeiT-Base model, this proportion increases to 45.1%. Moreover, the percentage of FFN layers' latency in the large-scale ViT-Large model rises to 53.8%, more than half of the total inference time.

This phenomenon arises because scaling up ViTs typically involves increasing the number of channels, whereas the number of tokens tends to remain constant. Meanwhile, the computational complexity of an FFN layer, quantified as $O(2\rho NC^2)$, is quadratic to the number of feature channels. Consequently, as the model expands, the FFN layers become significantly more computationally expensive. In conclusion, optimizing FFN layers becomes considerably important for minimizing the overall computational costs for large ViTs.

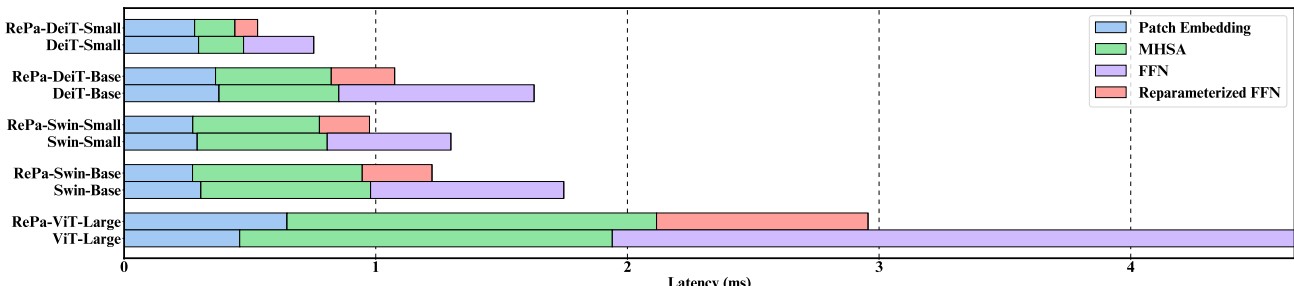

Figure 3. **Latency analysis.** Visualization of the runtime latencies of patch embedding, MHSA and FFN layers. Notably, as the model size increases, the proportion of latency attributed to FFN layers also rises. Our method effectively reduces the latency of FFN layers and obtains increasingly better performance on larger models, demonstrating a scalable acceleration of FFN layers.

## 3.2. Channel Idle Mechanism for FFN Layers

As Figure 1(a) illustrates, a typical FFN layer consists of two linear projections with a nonlinear activation function in between. Given an input $X \in \mathbb{R}^{N \times C}$ where $N$ represents the number of tokens and $C$ denotes the number of feature channels, the FFN layer process can be formulated as

$$Y = \mathrm{FFN}(\mathrm{LN}(X)) + X = \mathrm{Act}(\mathrm{LN}(X)W^{\mathrm{In}})W^{\mathrm{Out}} + X, \quad (1)$$

where $W^{\mathrm{In}} \in \mathbb{R}^{C \times \rho C}$, $W^{\mathrm{Out}} \in \mathbb{R}^{\rho C \times C}$ are the linear projection weights, $\mathrm{LN}(\cdot)$ is LayerNorm (Lei Ba et al., 2016) and $\mathrm{Act}(\cdot)$ is usually the GELU (Hendrycks & Gimpel, 2016) activation function. $\rho$ is the FFN expansion ratio, which is usually set to 4. The biases are omitted for simplicity since they are inherently linear and do not interfere with the reparameterization process. Unfortunately, due to the nonlinear activation function, the structural reparameterization cannot directly merge the two linear projection weights $W^{\mathrm{In}}$ and $W^{\mathrm{Out}}$ via linear algebra operations.

Inspired by ShuffleNetv2 (Ma et al., 2018) which keeps a group of channels idle in grouped convolutions and shuffles channels for information exchange, we propose a simple yet effective channel idle mechanism to enable reparameterization in FFN layers. Specifically, this mechanism maintains a large subset of feature channels inactivated in an FFN layer, thereby bridging a linear pathway through the nonlinear activation function in the corresponding FFN layer. In addition, we substitute LayerNorm with BatchNorm (BN) (Ioffe & Szegedy, 2015) to enable post-training reparameterization of normalization and shortcut for the FFN layer. As a result, our channel idle mechanism during the training stage can be formulated as

$$\begin{aligned} X^{\mathrm{In}} &= \mathrm{BN}(X)W^{\mathrm{In}}, \\ X^{\mathrm{Act}} &= \mathrm{Concat}(\mathrm{Act}(X^{\mathrm{In}}_{[:, \, 1:\mu C]}), X^{\mathrm{In}}_{[:, \, \mu C+1:\rho C]}), \quad (2) \\ Y &= \mathrm{BN}(X^{\mathrm{Act}})W^{\mathrm{Out}} + X, \end{aligned}$$

where the activation function is only applied on $\mu C$ ($\mu < \rho$) feature channels. The $(\rho - \mu)C$ idling feature channels construct a linear route as presented in Figure 1(b).

We further define the channel idle ratio as $\theta = 1 - \frac{\mu}{\rho}$, which represents the percentage of feature channels keeping inactivated in the FFN layer. $\mu$ is set to 1 by default in the following experiments unless otherwise noted, leading to the default $\theta = 1 - \frac{1}{\rho}$ (*e.g.*, $\theta = 0.75$ when $\rho = 4$, indicating 75% channels are idling when the expansion ratio is 4).

## 3.3. Structural Reparameterization for FFN layers

With the channel idle mechanism defined in Equation 2, we are able to simplify the FFN layer by structural reparameterization during the testing stage. Firstly, we reparameterize the BatchNorms into their corresponding linear projection weights as

$$\begin{aligned} \widetilde{W}^{\mathrm{In}} &= \frac{\gamma_X}{\sqrt{\sigma_X^2 + \epsilon_X}} W^{\mathrm{In}}, \\ \widetilde{W}^{\mathrm{Out}} &= \frac{\gamma_{X^{\mathrm{Act}}}}{\sqrt{\sigma_{X^{\mathrm{Act}}}^2 + \epsilon_{X^{\mathrm{Act}}}}} W^{\mathrm{Out}}, \end{aligned} \quad (3)$$

where $\gamma$s, $\sigma^2$s and $\epsilon$s are the empirical means, empirical variances and constants from the frozen BatchNorm layers, respectively. With the reparameterized projection weights $\widetilde{W}^{\mathrm{In}}$ and $\widetilde{W}^{\mathrm{Out}}$, the output $Y$ in Equation 2 can be reformulated as

$$\begin{aligned} Y = &\mathrm{Act}(X\widetilde{W}^{\mathrm{In}}_{[:, \, 1:\mu C]})\widetilde{W}^{\mathrm{Out}}_{[1:\mu C, \, :]} \\ &+ X\widetilde{W}^{\mathrm{In}}_{[:, \, \mu C+1:\rho C]}\widetilde{W}^{\mathrm{Out}}_{[\mu C+1:\rho C, \, :]} + X. \end{aligned} \quad (4)$$

Then, we further reparameterize the weights as

$$\widetilde{W} = \widetilde{W}^{\mathrm{In}}_{[:, \, \mu C+1:\rho C]}\widetilde{W}^{\mathrm{Out}}_{[\mu C+1:\rho C, \, :]} + I. \quad (5)$$

By substituting Equation 5 into Equation 4, we obtain the updating function for the FFN layer during the testing stage with three reparameterized weights as

$$Z = \mathrm{Act}(Y\widetilde{W}^{\mathrm{In}}_{[:, \, 1:\mu C]})\widetilde{W}^{\mathrm{Out}}_{[1:\mu C, \, :]} + Y\widetilde{W}. \quad (6)$$

As Figure 1(c) shows, after reparameterization, the two massive linear projections are converted into three smaller linear transformations with fewer parameters and all the normalizations are merged into linear projection weights.

## 3.4. Computational Complexity Analysis

**Number of parameters:** The vanilla FFN layer's parameters are mainly derived from the two linear projection weights $W^{\text{In}} \in \mathbb{R}^{C \times \rho C}$ and $W^{\text{Out}} \in \mathbb{R}^{\rho C \times C}$, totalling $2\rho C^2$. In contrast, with our channel idle mechanism, the weights are reparameterized into three terms: an input weight $\widetilde{W}^{\text{In}}_{[:,\ 1:\mu C]} \in \mathbb{R}^{C \times \mu C}$, an output weight $\widetilde{W}^{\text{Out}}_{[1:\mu C,\ :]} \in \mathbb{R}^{\mu C \times C}$ and a reparameterized weight $\widetilde{W} \in \mathbb{R}^{C \times C}$. The total number of parameters is effectively reduced from $2\rho C^2$ to $(2\mu + 1)C^2$.

Consequently, in the reparameterized FFN layer, the parameter count is diminished to $1 - \theta + \frac{1}{2\rho}$ of the original parameter count, where $\theta$ is the aforementioned idle ratio. For instance, when $\rho = 4$ and $\theta = 0.75$, the number of parameters in an FFN layer declines to 37.5% post-parameterization. This reduction significantly simplifies the model, diminishing its memory consumption.

**Computational complexity:** The computational complexity of the vanilla FFN layer is $O(2\rho NC^2)$ while the computational complexity is significantly reduced to $O((2\mu + 1)NC^2)$ in our reparameterized FFN layer. The computational complexity reduction ratio for an FFN layer is also $1 - \theta + \frac{1}{2\rho}$.

It is worth noting that, due to the elimination of normalizations and shortcuts in the FFN layer, the inference speed gain is more than the computational complexity reduction.

## 3.5. Comparison against RepVGG-style Reparameterization

RepVGG (Ding et al., 2021b) introduces structural reparameterization into CNNs, where multi-branch convolutions are merged into a single-branch convolution through linear operations on convolution kernels. While RePaViT draws inspiration from RepVGG, there are significant differences between our structural reparameterization approach and the RepVGG-style reparameterization:

- **Different targets:** Existing works using RepVGG-style reparameterization for efficient ViTs (Vasu et al., 2023a;b) introduce CNN components into ViTs and only reparameterize those convolutional components. In contrast, our method directly targets existing FFN layers in ViTs, aiming to improve the efficiency of standard ViT architectures rather than designing an entirely new backbone. Thus, the application objectives are fundamentally distinct.

- **Different reparameterization solutions:** Another difference is that RepVGG reparameterizes **horizontally** across parallel convolutional kernels, while RePaViT reparameterizes **vertically** on consecutive linear projection weights. Mathematically, RepVGG reparameterizes two parallel convolutional branches with kernels $W_1^{\text{Conv}}$ and $W_2^{\text{Conv}}$ by

summing them:

$$\widetilde{W}^{\text{Conv}}_{\text{Rep}} = W_1^{\text{Conv}} + W_2^{\text{Conv}}. \tag{7}$$

On the contrary, as demonstrated in Equation 5, RePaViT reparameterizes two consecutive projection weights $W_1^{\text{FFN}}$ and $W_2^{\text{FFN}}$ by multiplying them:

$$\widetilde{W}^{\text{FFN}}_{\text{Rep}} = W_1^{\text{FFN}} \cdot W_2^{\text{FFN}}. \tag{8}$$

In the above example, $W_1^{\text{Conv}}$ and $W_2^{\text{Conv}}$ have been padded to the same shape, and the reparameterization processes of BatchNorm and biases are omitted for simplicity.

It is also worth noting that our channel idle mechanism cannot be regarded as a special case of a dual-branch structure in RepVGG. In RepVGG, all branches must be linear so that they can be reparameterized, whereas in our approach, one branch is linear while the other one is nonlinear.

# 4. Experiments

## 4.1. Datasets, Training and Evaluation Settings

We mainly train and test RePaViTs for the image classification task on the widely recognized ImageNet-1k (Deng et al., 2009) dataset, following the data augmentations and training recipes proposed by Touvron et al. (2021) as the standard practice. In line with Yao et al. (2021), the maximum learning rate is set to $4 \times 10^{-3}$ with 20 epochs of warmup from $1 \times 10^{-6}$. The default batch size and total training epochs are 4096 and 300, respectively. For dense prediction tasks, we follow the configurations from MMDetection (Chen et al., 2019) and MMSegmentation (Contributors, 2020) to finetune RePaViTs on MSCOCO (Lin et al., 2014) and ADE20K (Zhou et al., 2017) datasets for object detection and segmentation tasks, respectively. All the models are trained from scratch on NVIDIA H100 GPUs. To ensure fair comparisons, we measure the throughput of all the models on the same NVIDIA A6000 GPU with the same environments and a fixed batch size of 128. FlashAttention (Dao et al., 2022) is used for self-attention computation during inference measurement by default. More implementation details on the training settings are provided in Appendix A.

## 4.2. Classification Results

**Backbones:** We choose four ViT backbones, including a representative plain-structured ViT (DeiT (Touvron et al., 2021)), a representative hierarchical-structured ViT (Swin Transformer (Liu et al., 2021)), a plain ViT trained with token labelling (LV-ViT (Jiang et al., 2021)), and large-scale ViT (Dosovitskiy et al., 2021). The FFN layers in these models are embedded with the channel idle mechanism and are all trained from scratch solely on the ImageNet-1k dataset by supervised learning.

*Table 1.* **Performance comparisons among RePaViTs and their vanilla backbones.** For the "RePa" column, × and √ stands for the RePaViT model pre- and post-reparameterization, respectively. The decimals after model names (*i.e.*, 0.50 and 0.75) represent the channel idle ratios ($\theta$). When the backbone architecture fixes, our method consistently achieves greater accelerations and complexity reductions while narrowing the accuracy gap as the model size grows.

| Model | RePa | #MParam. ↓ | Complexity (GMACs) ↓ | Speed (images/second) ↑ | Top-1 accuracy ↑ |
|---|---|---|---|---|---|
| DeiT-Tiny | - | 5.7 | 1.1 | 3435.1 | 72.1% |
| RePa-DeiT-Tiny/0.50 | × | 5.7 | 1.1 | 2397.9 | 69.4% (−2.7%) |
| | √ | 4.4 (−22.8%) | 0.8 (−27.3%) | 4001.2 (+16.5%) | |
| DeiT-Small | - | 22.1 | 4.3 | 1410.3 | 79.8% |
| RePa-DeiT-Small/0.5 | × | 22.1 | 4.3 | 1000.9 | 78.9% (−0.9%) |
| | √ | 16.7 (−24.4%) | 3.2 (−25.6%) | 1734.7 (+23.0%) | |
| DeiT-Base | - | 86.6 | 16.9 | 418.5 | 81.8% |
| RePa-DeiT-Base/0.75 | × | 86.6 | 16.9 | 336.6 | 81.3% (−0.5%) |
| | √ | 51.1 (−41.0%) | 9.9 (−41.4%) | 660.3 (+57.8%) | |
| ViT-Large | - | 304.3 | 59.7 | 124.2 | 80.3% |
| RePa-ViT-Large/0.75 | × | 304.5 | 59.8 | 102.7 | 82.0% (+1.7%) |
| | √ | 178.4 (−41.4%) | 34.9 (−41.5%) | 207.2 (+66.8%) | |
| ViT-Huge | - | 632.2 | 124.3 | 61.5 | 80.3% |
| RePa-ViT-Huge/0.75 | × | 632.5 | 124.4 | 53.0 | 81.4% (+1.1%) |
| | √ | 369.9 (−41.5%) | 72.6 (−41.6%) | 103.8 (+68.7%) | |
| Swin-Tiny | - | 28.3 | 4.4 | 804.4 | 81.2% |
| RePa-Swin-Tiny/0.75 | × | 28.3 | 4.4 | 614.9 | 78.4% (−2.8%) |
| | √ | 17.5 (−38.2%) | 2.6 (−40.9%) | 1020.4 (+26.9%) | |
| Swin-Small | - | 49.6 | 8.6 | 471.7 | 83.0% |
| RePa-Swin-Small/0.75 | × | 49.7 | 8.6 | 363.1 | 81.4% (−1.6%) |
| | √ | 29.9 (−39.7%) | 5.1 (−40.7%) | 627.8 (+33.1%) | |
| Swin-Base | - | 87.8 | 15.2 | 326.6 | 83.5% |
| RePa-Swin-Base/0.75 | × | 87.9 | 15.2 | 249.4 | 82.6% (−0.9%) |
| | √ | 52.8 (−39.9%) | 9.0 (−40.8%) | 467.6 (+43.2%) | |
| LV-ViT-S | - | 26.2 | 6.1 | 866.6 | 81.4% |
| RePa-LV-ViT-S/0.75 | × | 26.2 | 6.1 | 725.4 | 81.6% (+0.2%) |
| | √ | 19.1 (−27.1%) | 4.7 (−23.0%) | 1110.9 (+28.2%) | |
| LV-ViT-M | - | 55.8 | 11.9 | 457.6 | 83.6% |
| RePa-LV-ViT-M/0.75 | × | 55.9 | 11.9 | 396.6 | 83.5% (−0.1%) |
| | √ | 40.1 (−28.1%) | 8.8 (−26.1%) | 640.6 (+40.0%) | |

*Table 2.* **Comparison with state-of-the-art network pruning methods for efficient ViTs.** "-" indicates that the statistic is either missing or irreproducible. Our method demonstrates significantly higher speed-ups compared to pruning methods while achieving competitive or even higher top-1 accuracies across various ViT backbones.

| Backbone | Method | #MParam. ↓ | Compl. (GMACs) ↓ | Speed improv. ↑ | Top-1 acc. ↑ |
|---|---|---|---|---|---|
| DeiT-Small | WDPruning | 13.3 | 2.6 | +18.3% | 78.4% |
| | X-pruner | - | 2.4 | - | 78.9% |
| | DC-ViT | 16.6 | 3.2 | +20.0% | 78.6% |
| | LPViT | 22.1 | **2.3** | +16.3% | **80.7%** |
| | RePaViT/0.50 | 16.7 | 3.2 | +23.0% | 78.9% |
| | RePaViT/0.75 | **13.2** | 2.5 | **+42.1%** | 77.0% |
| DeiT-Base | WDPruning | 55.3 | 9.9 | +18.2% | 80.8% |
| | X-pruner | - | 8.5 | - | 81.0% |
| | DC-ViT | 65.1 | 12.7 | +18.4% | 81.3% |
| | LPViT | 86.6 | 8.8 | +18.8% | 80.8% |
| | RePaViT/0.50 | 65.3 | 12.7 | +28.6% | **81.4%** |
| | RePaViT/0.75 | **51.1** | 10.6 | **+57.8%** | 81.3% |
| Swin-Small | WDPruning | 32.8 | 6.3 | +15.3% | 81.8% |
| | X-pruner | - | 6.0 | - | 82.0% |
| | RePaViT/0.50 | 37.8 | 6.4 | +20.7% | **82.8%** |
| | RePaViT/0.75 | **29.9** | **5.1** | **+33.1%** | 81.4% |
| Swin-Base | DC-ViT | 66.4 | 11.5 | +14.9% | **83.8%** |
| | LPViT | 87.8 | 11.2 | +8.9% | 81.7% |
| | RePaViT/0.50 | 66.8 | 11.5 | +19.6% | 83.4% |
| | RePaViT/0.75 | **52.8** | **9.0** | **+42.4%** | 82.6% |

*Table 3.* **Comparison against the state-of-the-art reparameterization method for ViTs.** With a similar number of parameters, RePaViT obtains both faster inference speeds and higher accuracies than SLAB (Guo et al., 2024).

| Model | #MParam. ↓ | Compl. (GMACs) ↓ | Speed (img/s) ↑ | Top-1 acc. ↑ |
|---|---|---|---|---|
| SLAB-DeiT-Base | 86.6 | 17.1 | 387.0 | 78.9% |
| RePa-DeiT-Base/0.25 | **79.5** | **15.5** | **452.3** | **81.1%** |
| SLAB-Swin-Base | 87.7 | 15.4 | 299.9 | 83.6% |
| RePa-Swin-Base/0.25 | **80.8** | **14.0** | **356.3** | **83.7%** |

**Reparameterization results:** Table 1 presents the image classification performance of RePaViTs before and after reparameterization, and compares with their vanilla backbones. Due to the nature of linear algebra operations, the pre- and post-reparameterization accuracies are the same.

In general, our innovative channel idle mechanism remarkably enhances these models' computational efficiency and throughput while preserving their accuracy. We observe that **with the same backbone architecture, RePaViT achieves more substantial acceleration with a narrowing accuracy gap when the model size increases.** For example, employing DeiT as the backbone, the smaller DeiT-Tiny model witnesses a 16.5% speed-up at the cost of a 2.7% accuracy loss. However, when scaled up to the DeiT-Base model, our approach delivers a 57.8% throughput improvement, with only a marginal 0.5% drop in accuracy. This pattern is consistent across various models. In cases where the backbones include additional regularizations during training, our method not only accelerates performance but also preserves accuracy to a remarkable extent. In particular, on the LV-ViT-M model, we facilitate a 40.0% increase in the inference speed with a negligible 0.1% decrease in accuracy.

Notably, **RePaViT yields ~68% speed-up and even 1~2% higher accuracy on ViT-Large and ViT-Huge models**, indicating its potential on large-scale foundation models. This insight demonstrates the practical value of RePaViT in accelerating large-scale models without compromising performance, making it an effective solution for large-scale real-world applications requiring both speed and precision.

### 4.3. Comparison Against Network Pruning

While several network pruning methods for efficient ViTs focus on reducing the number of parameters and the theoretical computational complexity during inference, our approach differs fundamentally from these methods. We provide a comparison with state-of-the-art and representative network pruning techniques in Table 2, including WDPruning (Yu et al., 2022a), X-Pruner (Yu & Xiang, 2023), DC-ViT

*Table 4.* **Sensitivity of channel idle ratio** $\theta$. The performance of RePaViT on plain (DeiT (Touvron et al., 2021)) and hierarchical (Swin (Liu et al., 2021)) ViTs with various $\theta$ is reported. $\theta$=* represents the vanilla backbone. $\theta$=1.00 implies the nonlinear activation being removed from the model. The results show a significant accuracy drop when $\theta$ surpasses 0.75.

| Backbone | Idle ratio $\theta$ | #MParam. ↓ | Compl. (GMACs) ↓ | Speed (img/s) ↑ | Top-1 acc. ↑ |
|---|---|---|---|---|---|
| DeiT-Tiny | 1.00 | 2.6 | 0.5 | 5810.1 | 48.6% |
| | 0.75 | 3.5 | 0.6 | 4470.8 | 64.2% |
| | 0.50 | 4.4 | 0.8 | 4001.2 | 69.4% |
| | 0.25 | 5.3 | 1.0 | 3575.6 | 71.9% |
| | * | 5.7 | 1.1 | 3435.1 | **72.1%** |
| DeiT-Small | 1.00 | 9.6 | 1.8 | 2612.9 | 63.9% |
| | 0.75 | 13.2 | 2.5 | 2003.7 | 77.0% |
| | 0.50 | 16.7 | 3.2 | 1734.7 | 78.9% |
| | 0.25 | 20.3 | 3.9 | 1489.7 | **80.3%** |
| | * | 22.1 | 4.3 | 1410.3 | 79.8% |
| DeiT-Base | 1.00 | 37.0 | 7.1 | 878.7 | 73.7% |
| | 0.75 | 51.1 | 9.9 | 660.3 | 81.3% |
| | 0.50 | 65.3 | 12.7 | 538.0 | 81.4% |
| | 0.25 | 79.5 | 15.5 | 452.3 | 81.1% |
| | * | 86.6 | 16.9 | 418.5 | **81.8%** |
| Swin-Tiny | 1.00 | 13.2 | 1.9 | 1180.1 | 67.6% |
| | 0.75 | 17.5 | 2.6 | 1020.4 | 78.4% |
| | 0.50 | 21.8 | 3.3 | 905.9 | 80.5% |
| | 0.25 | 26.1 | 4.0 | 844.8 | **81.4%** |
| | * | 28.3 | 4.4 | 804.4 | 81.2% |
| Swin-Small | 1.00 | 22.1 | 3.7 | 745.0 | 72.5% |
| | 0.75 | 29.9 | 5.1 | 627.8 | 81.4% |
| | 0.50 | 37.8 | 6.5 | 569.2 | 82.8% |
| | 0.25 | 45.7 | 7.9 | 514.5 | **83.1%** |
| | * | 49.6 | 8.6 | 471.7 | 83.0% |
| Swin-Base | 1.00 | 38.8 | 6.5 | 539.0 | 75.5% |
| | 0.75 | 52.8 | 9.0 | 467.6 | 82.6% |
| | 0.50 | 66.8 | 11.5 | 390.6 | 83.4% |
| | 0.25 | 80.8 | 14.0 | 356.3 | **83.7%** |
| | * | 87.8 | 15.2 | 326.6 | 83.5% |

*Table 5.* **Ablation study on train-time reparameterization.** $\sqrt{}$ for "Training RePa" stands for reparameterizing the model before training. $\sqrt{}$ for "BatchNorm RePa" represents that the Batch-Norm before a linear projection is reparameterized into the projection weight. "-" under top-1 accuracy means training failure. Overall, training with full parameters and reparameterizing during testing yields better performance.

| Model | Training RePa | BatchNorm RePa | Training #MParam. | Top-1 accuracy ↑ |
|---|---|---|---|---|
| RePa-DeiT-Tiny/0.75 | $\sqrt{}$ | $\sqrt{}$ | 3.5 | 59.6% |
| | $\sqrt{}$ | × | 3.5 | **64.3%** |
| | × | × | 5.7 | 64.2% |
| RePa-DeiT-Small/0.75 | $\sqrt{}$ | $\sqrt{}$ | 13.2 | 75.0% |
| | $\sqrt{}$ | × | 13.2 | 75.7% |
| | × | × | 22.1 | **77.0%** |
| RePa-DeiT-Base/0.75 | $\sqrt{}$ | $\sqrt{}$ | 51.1 | - |
| | $\sqrt{}$ | × | 51.1 | 80.6% |
| | × | × | 86.6 | **81.3%** |
| RePa-ViT-Large/0.75 | $\sqrt{}$ | $\sqrt{}$ | 178.4 | - |
| | $\sqrt{}$ | × | 178.5 | 80.6% |
| | × | × | 304.5 | **82.0%** |
| RePa-Swin-Tiny/0.75 | $\sqrt{}$ | $\sqrt{}$ | 17.5 | 77.1% |
| | $\sqrt{}$ | × | 17.5 | 78.0% |
| | × | × | 28.3 | **78.4%** |
| RePa-Swin-Small/0.75 | $\sqrt{}$ | $\sqrt{}$ | 29.9 | 79.3% |
| | $\sqrt{}$ | × | 30.0 | 79.1% |
| | × | × | 49.7 | **81.4%** |
| RePa-Swin-Base/0.75 | $\sqrt{}$ | $\sqrt{}$ | 52.8 | 79.6% |
| | $\sqrt{}$ | × | 52.9 | 80.3% |
| | × | × | 87.9 | **82.6%** |
| RePa-LV-ViT-S/0.75 | $\sqrt{}$ | $\sqrt{}$ | 19.1 | - |
| | $\sqrt{}$ | × | 19.1 | 81.3% |
| | × | × | 26.2 | **81.6%** |
| RePa-LV-ViT-M/0.75 | $\sqrt{}$ | $\sqrt{}$ | 40.1 | - |
| | $\sqrt{}$ | × | 40.2 | - |
| | × | × | 55.9 | **83.6%** |

(Zhang et al., 2024), and LPViT (Xu et al., 2024a). Due to unavailable or incomplete code repositories of certain state-of-the-art pruning methods, we rely on the performance statistics reported in the original papers and align efficiency optimization using speed improvements for fairness.

Table 2 shows that the structural reparameterization approach of RePaViT achieves significantly greater inference acceleration compared to network pruning methods. Moreover, the effectiveness of our method increases as model size grows. For example, while the state-of-the-art DC-ViT achieves speed improvements of approximately 15~20% across all backbones, RePaViT provides 19.6% to 57.8% speed improvements when the model scales up. These results highlight two key advantages of our method:

• **Computing environment friendly**: Our reparameterized model is dense and structurally regular, making it efficient to run on general-purpose hardware without requiring spe-cialized hardware and software support for sparse matrix operations. So our method can bring more speed-ups in general computing environments.

• **Scaling effectiveness on larger models**: Compared with network pruning methods, RePaVit yields more accelerations and smaller performance gaps on larger models even with the same channel idle ratio $\theta$. This underscores the important practical value of RePaViT on large foundation models for vision tasks.

### 4.4. Comparison Against State-of-The-Art Method

Table 3 compares our RePaViT approach against SLAB (Guo et al., 2024), a recent state-of-the-art method introducing progressive reparameterized BatchNorms for ViTs. For fair comparisons with similar model sizes, the performance of RePaViTs with $\theta$=0.25 is used. The results indicate that our reparameterization strategy offers a better trade-off be-

*Table 6.* **Performance on dense prediction tasks.** Results on the $1\times$ training schedule are presented. The latencies (ms) per image are reported for throughput comparisons.

| Model | RetinaNet | | | | | | | Mask R-CNN | | | | | | | UperNet | |
|---|---|---|---|---|---|---|---|---|---|---|---|---|---|---|---|---|
| | Latency (ms) ↓ | AP↑ | AP$_{50}$↑ | AP$_{75}$↑ | AP$_S$↑ | AP$_M$↑ | AP$_L$↑ | Latency (ms) ↓ | AP↑ | AP$_{50}$↑ | AP$_{75}$↑ | AP$_S$↑ | AP$_M$↑ | AP$_L$↑ | Latency (ms) ↓ | mIoU↑ |
| Swin-Small | 61.7 | 37.2 | 56.9 | 39.6 | **22.4** | 40.5 | 49.4 | 62.5 | **45.5** | 67.8 | **49.9** | 28.6 | 49.2 | 60.4 | 36.3 | **47.6** |
| RePa-Swin-Small | **53.8** (−12.8%) | **38.3** | **57.9** | **40.7** | 21.8 | **42.0** | **51.6** | **53.8** (−13.9%) | 43.6 | 65.8 | 47.8 | 27.1 | 47.0 | 57.3 | **32.1** (−11.6%) | 45.7 |
| Swin-Base | 82.0 | 38.9 | 59.5 | 41.3 | 24.3 | 43.6 | **54.4** | 82.6 | **45.8** | 67.6 | 50.3 | 28.7 | 48.9 | 61.7 | 45.6 | **48.1** |
| RePa-Swin-Base | **66.7** (−18.7%) | **39.8** | **60.0** | **42.1** | **25.3** | **43.7** | 53.8 | **69.4** (−16.0%) | 44.8 | 67.0 | 49.4 | **29.0** | 48.5 | 58.4 | **38.6** (−15.4%) | 46.9 |

tween efficiency and accuracy. For example, when utilizing DeiT-Base as the backbone, our method not only achieves a higher speed and fewer parameters but also surpasses SLAB by a 2.2% higher accuracy.

### 4.5. Sensitivty of Channel Idle Ratio $\theta$

In Section 3.2, we define the channel idle ratio $\theta$ as the percentage of feature channels keeping idle in the activation. Table 4 illustrates the influence of $\theta$ on the performance of RePaViTs. Overall, a larger $\theta$ represents more channels idling in the FFN layer, leading to a smaller number of parameters, a lower computational complexity, and a higher inference speed post-reparameterization.

Remarkably, when $\theta$ exceeds 0.75, which is the default idle ratio for RePaViTs, there is an obvious decline in the top-1 accuracies. For instance, when setting $\theta$ to 1.0 (*i.e.*, no channels being activated), the RePa-DeiT-Base's accuracy drops from 81.8% to 73.7%. Similarly, the RePa-Swin-Base model witnesses its accuracy decline from 83.5% to 75.5% with $\theta = 1.0$. For smaller models, such performance collapse can be more severe. This outcome demonstrates that while reducing the proportion of nonlinear components can significantly enhance the model's efficiency, preserving sufficient nonlinearities is also crucial for performance.

It is noteworthy that, with a proper $\theta$, ViTs can achieve even better performance with fewer parameters and faster inference speeds. For example, DeiT-Small, Swin-Tiny, Swin-Small and Swin-Base models all enjoy higher top-1 accuracy when $\theta$=0.25.

### 4.6. Ablation Study

We ablate the structural reparameterization process during training. Instead of training the full $2\rho C^2$ linear project weights and then reparameterizing them during testing, we directly train the reparameterized weights with a reduced size of $(2\mu + 1)C^2$. Specifically, in our experiments, the numbers of parameters for a single FFN layer before and after reparameterization are $8C^2$ (*i.e.*, $\rho$=4) and $3C^2$ (*i.e.*, $\mu$=1), respectively. Table 5 indicates that training with more parameters (*i.e.*, train-time overparameterization) generally

achieves better performance than training with less parameters for ViTs, which aligns with the findings in Vasu et al. (2023a;b). Meanwhile, train-time overparameterization also helps to stabilize the training process for large models. For instance, when trained with reparameterized structure, RePa-DeiT-Base, RePa-ViT-Large, RePa-LV-ViT-S and RePa-LV-ViT-M all suffer training collapse and fail to converge.

### 4.7. Dense Predictions

Table 6 presents the results of two downstream tasks. Firstly, the ImageNet-1k pre-trained RePa-Swin models are integrated with a one-stage detector RetinaNet (Lin et al., 2017) and a two-stage detector Mask R-CNN (He et al., 2017) for the object detection task on the MSCOCO dataset with $1\times$ training schedule (*i.e.*, 12 epochs). Remarkably, our RePa-Swin-Base model achieves up to 18.7% latency reduction at even a higher average precision (AP) with RetinaNet when compared to its vanilla backbone. RePA-Swin-Base also obtains a similar performance with 16.0% less latency with Mask R-CNN. Secondly, UperNet (Xiao et al., 2018) is leveraged for the semantic segmentation task on the ADE20K dataset with RePa-Swin models as backbones. Similarly, RePa-Swin-Base achieves 15.4% latency reduction with merely 1.2% mIoU loss.

Overall, the experimental results on downstream tasks reflect a consistent trend that the performance disparities are narrowing and the acceleration gains are escalating as the backbone model sizes grow. This aligns with the observations in Section 4.2 well, which further proves the scalable acceleration capability of our channel idle mechanism.

### 4.8. Self-supervised Learning Experiments and Others

Given that large foundation models are typically trained using self-supervised learning strategies, we evaluate RePaViT under self-supervised training (*i.e.*, DINO (Caron et al., 2021)) and language-guided contrastive learning (*i.e.*, CLIP (Radford et al., 2021)). The experimental results are provided in Appendix B. Notably, when applied to CLIP models, RePaViT improves zero-shot top-1 accuracy by 0.8% while achieving a 24.7% speed improvement, demonstrating its effectiveness in optimizing large foundation models.

# 5. Conclusion

In this paper, we investigate the latency compositions of ViTs and observe that FFN layers significantly contribute to the overall latency. The observations highlight the critical need for accelerating FFN layers to enhance the efficiency of ViTs, where structural reparameterization emerges as a potential solution. We introduce a novel channel idle mechanism to facilitate the reparameterization of FFN layers during inference. The proposed mechanism is employed on various ViT backbones, resulting in a family of RePaViTs. RePaViTs demonstrate consistent scalability with more accelerations and narrower accuracy disparities as the backbone model size escalates. Notably, RePaViT achieves accuracy gains while improving the inference speed on large-scale ViT backbones. These unprecedented results mark a disruptive and timely contribution to the community and establish RePaViT as a significant addition to the toolkit for accelerating large foundation models. We believe that RePaViT presents a promising direction for expediting ViTs and we invite the community to further explore its effectiveness on even larger foundation models.

# Impact Statement

This paper presents work whose goal is to advance the field of Machine Learning. There are many potential societal consequences of our work, none which we feel must be specifically highlighted here.

# Acknowledgement

This research was partially supported by the Australian Government through the Australian Research Council's Industrial Transformation Training Centre for Information Resilience (CIRES) project number IC200100022, CSIRO's Research Plus Science Leader Project R-91559, and Australian Research Council Discovery Projects DP230101753 and DECRA DE200101610.

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

## A. Training Settings

All RePaViTs are rigorously trained on the ImageNet-1k dataset (Deng et al., 2009), following the same data augmentations proposed by DeiT (Touvron et al., 2021). Consistently, the total number of training epochs is standardized at 300. In an effort to accommodate the substitution of LayerNorm with BatchNorm, we have increased the batch size to 4096. Additionally, the Lamb optimizer (You et al., 2020) has been selected to ensure stable training with a large batch size. Learning rates are dedicatedly configured for different backbone architectures, and a cosine scheduler (Loshchilov & Hutter, 2017) is utilized for learning rate adjustment throughout the training period. Detailed training settings are provided in Table 7.

*Table 7.* Training settings of RePaViTs for the image classification task.

| Model | Epochs | Batch size | Optimizer | Base learning rate | Min learning rate | Warmup learning rate | Scheduler | Weight decay | Drop path rate |
|---|---|---|---|---|---|---|---|---|---|
| RePa-DeiT-Tiny RePa-DeiT-Small RePa-DeiT-Base | 300 | 4096 | Lamb | $4 \times 10^{-3}$ | $5 \times 10^{-5}$ | $1 \times 10^{-6}$ | Cosine scheduler | 0.05 | 0.10 |
| RePa-ViT-Large RePa-ViT-Huge | | | | $1 \times 10^{-3}$ | | | | | 0.30 |
| RePa-Swin-Tiny RePa-Swin-Small RePa-Swin-Base | | | | $4 \times 10^{-3}$ | | | | | 0.10 |
| RePa-LV-ViT-S RePa-LV-ViT-M | | 1024 | | $1 \times 10^{-3}$ | $1 \times 10^{-5}$ | | | | |

## B. Self-Supervised Learning Performance

Large foundation models with superior performance are usually trained with self-supervised learning techniques. To demonstrate the potential applicability of RePaViT with self-supervised learning, we first validate our method using DINO (Caron et al., 2021) and report the performance in Table 8. We adopt the same training settings as outlined in DINO. Even with self-supervised learning, RePaViTs still exhibit substantial efficiency enhancement.

Notably, there is a consistent trend as observed in Section 4.2 that when the model size increases, our method yields greater speed improvements and a smaller accuracy gap. For example, RePa-ViT-Small achieves a 39.4% increase in speed (1779.6 image/second vs 1277.0 image/second) with a 2.6% drop in accuracy (74.4% vs 77.0%) when using a linear classifier. In the case of employing a larger backbone model, RePa-ViT-Base realizes a more significant acceleration of 57.2% (623.0 image/second vs 396.2 image/second) with a smaller accuracy loss of 1.2% (77.0% vs 78.2%). These results indicate a high adaptability of our RePaViT using different learning paradigms.

*Table 8.* **RePaViT performance on DINO models (Caron et al., 2021).**

| Model | #MParam. ↓ | Compl. (GMACs) ↓ | Speed (img/s) ↑ | $k$-NN top-1 acc. ↑ | Linear top-1 acc. ↑ |
|---|---|---|---|---|---|
| ViT-Small | 21.7 | 4.3 | 1277.0 | 72.8% | 77.0% |
| RePa-ViT-Small/0.75 | 12.8 (−41.1%) | 2.5 (−41.9%) | 1779.6 (+39.4%) | 69.6% | 74.4% |
| ViT-Base | 85.8 | 16.9 | 396.2 | 76.1% | 78.2% |
| RePa-ViT-Base/0.75 | 50.4 (−41.3%) | 9.9 (−41.4%) | 623.0 (+57.2%) | 74.1% | 77.0% |

Next, we evaluate RePaViT on a more advanced language-guided contrastive learning framework, specifically CLIP (Radford et al., 2021). We adopt the open-source OpenCLIP framework (Cherti et al., 2023) and train all models on the LAION-400M dataset (Schuhmann et al., 2021), with a total of 3B seen data points. All training configurations strictly follow the default settings of OpenCLIP. The zero-shot classification performance on the ImageNet-1K validation set is presented in Table 9.

For the smaller CLIP-ViT-B/32 model, our RePa-CLIP-ViT-B/32 achieves a 26.8% speed increase with a negligible 0.3% accuracy drop. On the larger CLIP-ViT-B/16 model, our method improves inference speed by 24.7% while achieving a 0.8% gain in zero-shot classification top-1 accuracy. These results demonstrate the effectiveness of RePaViT in enhancing the

*Table 9.* **RePaViT performance on CLIP models (Radford et al., 2021).** All the models are trained on LAION-400M dataset with 3B seen samples in total.

| Model | Idle ratio $\theta$ | #MParam. ↓ | Complexity (GFLOPs) ↓ | Speed (image/second) ↑ | Top-1 accuracy ↑ |
|---|---|---|---|---|---|
| CLIP-ViT-B/32 | - | 87.9 | 4.4 | 3860.2 | 57.1% |
| RePa-CLIP-ViT-B/32 | 0.50 | 66.6 (−24.2%) | 3.4 (−22.7%) | 4893.5 (+26.8%) | 56.8% (−0.3%) |
| RePa-CLIP-ViT-B/32 | 0.75 | 52.4 (−40.4%) | 2.6 (−40.9%) | 5812.3 (+50.6%) | 53.2% (−3.9%) |
| CLIP-ViT-B/16 | - | 86.2 | 17.6 | 824.2 | 62.7% |
| RePa-CLIP-ViT-B/16 | 0.50 | 64.9 (−24.7%) | 13.4 (−23.9%) | 1027.9 (+24.7%) | 63.5% (+0.8%) |
| RePa-CLIP-ViT-B/16 | 0.75 | 50.8 (−41.1%) | 10.6 (−39.8%) | 1161.5 (+40.9%) | 61.0% (−1.7%) |

efficiency of large foundation models trained with language-guided contrastive learning. We anticipate our method to be applied to large foundational vision models in future work.

## C. Limitations

Despite the exceptional performance of RePaFormers on large backbone models, there is a notable decrease in accuracy as the model size shrinks. For example, as demonstrated in Table 4, the accuracy of RePa-DeiT-Tiny decreases significantly from 72.1% to 64.2%. This performance drop is primarily attributed to the reduced nonlinearity in the backbone, which is a consequence of keeping channels idle. In smaller models, both the number of layers and the number of feature channels are limited, resulting in substantially fewer activated channels compared to larger models. After applying the channel idle mechanism with a high idle ratio (*e.g.*, 75%), tiny models would lack sufficient non-linear transformations. However, as the model size increases, both the number of layers and feature channels expand, enhancing the model's robustness and mitigating the impact of reduced nonlinearity.

In conclusion, while our method may not be optimally suited for tiny models, it significantly enhances the performance of large ViT models. We sincerely invite the research community to further investigate and validate the effectiveness of our approach on large foundational models, such as SAM (Kirillov et al., 2023) or GPT (Radford et al., 2019; Brown et al., 2020). This exploration could provide valuable insights into the scalability and adaptability of our method across various advanced computational frameworks.

