# OpenReview forum: "RePaViT: Scalable Vision Transformer Acceleration via Structural Reparameterization on Feedforward Network Layers"
_ICML.cc/2025/Conference — ICML 2025 poster_

### Official Review · Reviewer_8Sgy · 2025-03-05

**Overall Recommendation:** 4

**Summary:**

This paper proposes RePaViT, a method for accelerating Vision Transformers (ViTs) through structural reparameterization of the feedforward network layers. Specifically, this paper argues that the computation costs of FFN layers cannot be ignored, thus a structural reparameterization method on FFN layers are designed. By using the channel idle mechanism, the RePaViT reduces the computation complexity in the inference stage.

**Claims And Evidence:**

The paper's claims are supported by some evidence, particularly through performance comparisons and ablation studies. However, there are still some problems remain. The details are listed below:

Pros:

1. Table 1 shows that the RePaViT can reduce the number of parameters, therefore improve the inference speed.

2. Table 3 shows that the RePaViT has better inference speed and accuracy than the previous re-parameterization methods.

3. Ablation studies shows that importance of the idle ratio and the training time overparameterization.



Cons:
1. Table 2 tries to show the advantages of RePaViT comparing with pruning methods. However, the number of model parameters of most of pruning methods are not provided, which means it is hard to directly compare them with the RePaViT.

2. Most of baseline methods are proposed in 2021. More recent baselines may make the experimental results stronger.

**Essential References Not Discussed:**

N/A

**Experimental Designs Or Analyses:**

The experimental designs are valid to support the method.

**Methods And Evaluation Criteria:**

The proposed methods and evaluation criteria in the paper make sense for the problem or application at hand.

**Other Comments Or Suggestions:**

N/A

**Other Strengths And Weaknesses:**

N/A

**Questions For Authors:**

N/A

**Relation To Broader Scientific Literature:**

The key contribution of the paper is a novel re-parameterization methods on FFN layers, which has some relationship with RepVGG(Ding et al., 2021), and the authors analyzed the main differences with RepVGG in 3.5

**Theoretical Claims:**

This paper does not include theoretical proofs.

---

> ### Author Rebuttal · Authors · 2025-04-01
>
> We appreciate Reviewer 8Sgy's recognition of our method's high performance and would like to address the concerns raised:
>
> ---
>
> __C1: Table 2 tries to show the advantages of RePaViT comparing with pruning methods. However, the number of model parameters of most of pruning methods are not provided, which means it is hard to directly compare them with the RePaViT.__
>
> A1: We acknowledge the difficulty in directly comparing RePaViT with certain pruning methods due to the unavailability of parameter counts in their publications and/or unavailability of source code:
> 1. __X-pruner [1] does not report the number of parameters in its paper and has not released source code so far.__ So, we're unable to provide its number of model parameters.
> 2. __DC-ViT [2] does not report the number of parameters in its paper and had not provide its source code before the submission deadline of ICML__. As a result, we're unable to provide its number of model parameters in our submission. However, we would like to reproduce its code and report the updates in the table below. Notably, DC-ViT's model size and computational complexity depends on both the number of pruned blocks and a special MLP compression ratio. Unfortunately, the authors haven't released the MLP compression ratios for reproducing their reported results (except 0.751734186 for DeiT-Base) and searching for exactly the same ratios would require a search process exceeding the rebuttal phase. So we can only provide the number of parameters and complexity that yields the closest performance as reported.
> 3. __LPViT [3], while its source code is available, has not released the hardware benchmarking code so far.__ It still employs masking to simulate pruning without reducing the actual parameter count, which remains equivalent to the original model. Our investigation on the source code confirm this. And this is also why LPViT only reports sparsity in its paper.
>
> To facilitate a more comprehensive comparison, we have reproduced several pruning baselines using their latest released code. The results are summarized in the table below (italic for updated, bold for best).
>
> |**Backbone**|**Method**|**#MParam. ↓**|**Compl. (GMACs) ↓**|**Speed improv. ↑**|**Top-1 acc. ↑**|
> |-|-|-|-|-|-|
> |**DeiT-Small**|WDPruning|13.3|2.6|+18.3%|78.4%|
> || X-pruner|-|2.4|-|78.9%|
> || DC-ViT|_16.6_|_3.2_| +20.0%| 78.6%|
> || LPViT|_22.1_|__2.3__| +16.3%| **80.7%** |
> | | RePaViT/0.50 | 16.7 | 3.2| +34.9%              | 79.1%            |
> |                | RePaViT/0.75 | **13.2**       | 2.9                  | **+54.4%**          | 79.6%            |
> | **DeiT-Base**  | WDPruning    | 55.3           | 9.9                  | +18.2%              | 80.6%            |
> || X-pruner     | -              | **8.5**              | -                   | 81.0%            |
> |  | DC-ViT       | _65.1_         | _12.7_               | _+18.4%_            | 81.3%            |
> |   | LPViT        | _86.6_         | 8.8                  | +18.8%              | 80.8%            |
> |   | RePaViT/0.50 | 65.3           | 12.7                 | +21.8%              | **81.4%**        |
> |                | RePaViT/0.75 | **51.1**       | 10.6                 | **+67.5%**          | **81.4%**        |
> | **Swin-Small** | WDPruning    | 32.8           | 6.3                  | +15.3%              | 81.3%            |
> |                | X-pruner     | -              | 6.0                  | -                   | 82.0%            |
> |  | RePaViT/0.50 | 37.8           | 6.4                  | +28.8%              | **82.8%**        |
> |   | RePaViT/0.75 | **29.9**       | **5.1**              | **+41.2%**          | 81.6%            |
> | **Swin-Base**  | DC-ViT       | _66.4_         | _11.5_               | +14.9%              | **83.8%**        |
> |   | LPViT        | _87.8_         | 11.2                 | +8.9%               | 81.7%            |
> | | RePaViT/0.50 | 66.8           | 11.5                 | +24.5%              | 83.4%            |
> |  | RePaViT/0.75 | **52.8**       | **9.0**              | **+49.6%**          | 82.6%            |
>
> We would like to emphasize that RePaViT still achieves the best trade-off in terms of model efficiency and accuracy.
>
> ---
>
> __C2: Most of baseline methods are proposed in 2021. More recent baselines may make the experimental results stronger.__
>
> A2: We respectfully disagree with this comment. All baseline methods compared in our study are published after 2021 as shown below:
>
> | Method    | Conference | Year |
> | --------- | ---------- | ---- |
> | WDPruning | AAAI       | 2022 |
> | X-Pruner  | CVPR       | 2023 |
> | DC-ViT    | CVPR       | 2024 |
> | LPViT     | ECCV       | 2024 |
> | SLAB      | ICML       | 2024 |
>
> And the backbones choices (DeiT/ViT/Swin/LV-ViT) follow the convention in token pruning and network pruning works.
>
> ---
>
> We sincerely appreciate your review comments, hope our responses satisfactorily address your concerns, and kindly request a consideration of recommendation score.

---

> > ### Comment · Reviewer_8Sgy · 2025-04-07
> >
> > C1: The authors' responses can solve my concern.
> > C2: My comment "Most of baseline methods are proposed in 2021" is corresponding to the Table 1, and I means that the authors may consider to apply the proposed method on some more recent backbone networks.
> >
> > Based on authors' rebuttal, I would like to keep my score unchange.

---

> > > ### Author Response · Authors · 2025-04-07
> > >
> > > We thank Reviewer 8Sgy for acknowledging our rebuttal responses and are glad to see your major concern on the baseline comparisons has been addressed. We also thank you for clarifying the suggestion on incorporating recent _"backbones"_, rather than _"baseline methods"_. Unfortunately, we do not have enough time to provide additional experiments on new backbones before the end of this discussion period. However, we respect your suggestion and are willing to include results on more recent backbones in the revised version when possible. Also, we will make our best effort to support more architectures in our released code.
> > >
> > > Regarding the backbone selections, our method proves to work well with DeiT/ViT/Swin/LV-ViT, which are also the prominent choices in recent state-of-the-art methods for token pruning and network pruning. This is because these are the most widely used ViT architectures in today's VFM and VLM families [1-4]. We would like to point out that the coverage of these architectures in our experiment provides a clear indication of this work's impact and relevance.
> > >
> > > The followings summarize the ViT backbones utilized in recently published works.
> > >
> > > Among the methods we compared in our paper:
> > >
> > > * __WDPruning__ [5]: DeiT-T/S/B and Swin-S.
> > >
> > > * __X-Pruner__ [6]: DeiT-T/S/B and Swin-T/S.
> > >
> > > * __DC-ViT__ [7]: ViT-T/S/B/L, DeiT-B and Swin-B.
> > >
> > > * __LPViT__ [8]: DeiT-S/B and Swin-T/B.
> > >
> > > * __SLAB__ [9]: DeiT-T/S, Swin-T/S/B and PVT-T/S/B.
> > >
> > > Some other recent network pruning works for ViTs:
> > >
> > > * __DIMAP__ [10]: Swin-T/S/B.
> > >
> > > * __NViT__ [11]: DeiT-T/S/B and Swin-S.
> > >
> > > And recent token pruning/merging works:
> > >
> > > * __TokenReduction__ [12]: DeiT-T/B.
> > >
> > > * __LTMP__ [13]: DeiT-T/S/B.
> > >
> > > * __ToMe__ [14]: DeiT-S and ViT-T/S/B/L/H.
> > >
> > > * __Nose__ [15]: DeiT-B.
> > >
> > > * __Zero-TPrune__ [16]: DeiT-T/S/B and LV-ViT-S.
> > >
> > > * __ToFu__ [17]: ViT-B/L and DeiT-S
> > >
> > > As listed above, recent works still use backbones introduced as early as 2021. In our paper, we have adopted ViT-L/H, DeiT-T/S/B, Swin-T/S/B and LV-ViT-S/M, which already cover a diverse range of small-to-large and plain-to-hierarchical ViT architectures.
> > >
> > > __Once again, we sincerely appreciate your recognition of our work and your insightful suggestions. We hope that our detailed explanation regarding the backbone selections will earn you stronger support.__
> > > \
> > > \
> > > \
> > > \
> > > __References__
> > >
> > > [1] Radford, Alec, et al. "Learning transferable visual models from natural language supervision." ICML, 2021.
> > >
> > > [2] Li, Liunian Harold, et al. "Grounded language-image pre-training." CVPR, 2022.
> > >
> > > [3] Kwon, Gukyeong, et al. "Masked vision and language modeling for multi-modal representation learning." ICLR, 2023.
> > >
> > > [4] Lin, Ji, et al. "Vila: On pre-training for visual language models." CVPR, 2024.
> > >
> > > [5] Yu, Fang, et al. "Width & depth pruning for vision transformers." AAAI, 2022.
> > >
> > > [6] Yu, Lu, and Wei Xiang. "X-pruner: explainable pruning for vision transformers." CVPR, 2023.
> > >
> > > [7] Zhang, Hanxiao, et al.  "Dense vision transformer compression with few samples." CVPR, 2024.
> > >
> > > [8] Xu, Kaixin, et al. "Lpvit: Low-power semi-structured pruning for vision transformers." ECCV, 2024.
> > >
> > > [9] Guo, Jialong, et al. "Slab: Efficient transformers with simplified linear attention and progressive re-parameterized batch normalization." ICML, 2024.
> > >
> > > [10] He, Yang, and Joey Tianyi Zhou. "Data-independent module-aware pruning for hierarchical vision transformers." ICLR, 2024.
> > >
> > > [11] Yang, Huanrui, et al. "Global vision transformer pruning with hessian-aware saliency." CVPR, 2023.
> > >
> > > [12] Haurum, Joakim Bruslund, et al. "Which tokens to use? investigating token reduction in vision transformers." ICCV, 2023.
> > >
> > > [13] Bonnaerens, Maxim, and Joni Dambre. "Learned thresholds token merging and pruning for vision transformers." TMLR, 2023.
> > >
> > > [14] Bolya, Daniel, et al. "Token merging: Your vit but faster." ICLR, 2023.
> > >
> > > [15] Lin, Sihao, et al. "Mlp can be a good transformer learner." CVPR, 2024.
> > >
> > > [16] Wang, Hongjie, et al. "Zero-TPrune: Zero-shot token pruning through leveraging of the attention graph in pre-trained transformers." CVPR, 2024.
> > >
> > > [17] Kim, Minchul, et al. "Token fusion: Bridging the gap between token pruning and token merging." WACV, 2024.

---

### Official Review · Reviewer_d1Yn · 2025-03-14

**Overall Recommendation:** 3

**Summary:**

This paper proposes a novel structural reparameterization method -- RePaViT that targets the feedforward network (FFN) layers of Vision Transformers (ViTs) to accelerate inference. The key idea is a channel idle mechanism—during training, only a subset of FFN channels are activated (with the others kept “idle”), which creates a linear shortcut that can later be merged into a simplified, reparameterized structure at test time. This approach not only reduces computational complexity (both in terms of parameter count and FLOPs) but, in many cases, also improves or preserves accuracy.

**Claims And Evidence:**

Claim: FFN layers are the major bottleneck in ViTs, and optimizing them can yield significant latency reductions.
Evidence: The latency analysis shows that FFN layers account for a growing portion of inference time as model size increases.

Claim: The channel idle mechanism and subsequent reparameterization can reduce both parameters and computation while maintaining accuracy.
Evidence: Extensive experiments on various backbones (DeiT, Swin, LV-ViT, ViT-Large/Huge) illustrate that RePaViT achieves up to 68% speedup and, in some cases, even higher accuracy compared to the original models.

**Essential References Not Discussed:**

N/A

**Experimental Designs Or Analyses:**

Experiments:
The authors conduct comprehensive experiments across multiple ViT backbones (plain and hierarchical) on ImageNet-1K, as well as on downstream tasks like object detection (MS COCO) and semantic segmentation (ADE20K).
Analysis:
Detailed tables compare the original and reparameterized models in terms of speed, accuracy, and computational cost.

**Methods And Evaluation Criteria:**

Methods:
In each FFN layer, only a fraction of the channels pass through the nonlinear activation while the rest follow a linear path. During inference, the activated and idle branches are merged using structural reparameterization (with BatchNorm merging) to form a more compact FFN. The approach is applied on standard ViT architectures trained from scratch on ImageNet-1K, and evaluations span image classification, object detection, and semantic segmentation.
Evaluation:
The paper reports throughput (images per second), parameter counts, FLOPs, and top-1 accuracy on ImageNet-1K. It further compares performance on dense prediction tasks (MS COCO, ADE20K) and benchmarks against network pruning and alternative reparameterization methods (e.g., SLAB).

**Other Comments Or Suggestions:**

See questions.

**Other Strengths And Weaknesses:**

Strengths:
1. The visualization is clear and understandable.
2. The experiments are solid and comprehensive.
3. The reparameterized models achieve good efficiency and accuracy.

Weaknesses:
1. Need more discussion about edge devices or resource limited scenarios.
2. Need more discussion about combination with other efficiency methods.

**Questions For Authors:**

1. How does RePaViT perform in deployment scenarios on edge devices?
2. I am curious about the performance of the proposed method when combined with other efficiency methods (e.g., quantization, pruning).

**Relation To Broader Scientific Literature:**

Its focus on directly optimizing FFN layers through a novel channel idle mechanism distinguishes it from existing methods that mainly target attention layers or use pruning strategies.

**Theoretical Claims:**

I do not think there are any theoretical claims. Most of the designs are empirical based.

---

> ### Author Rebuttal · Authors · 2025-04-01
>
> We sincerely appreciate Reviewer d1Yn's detailed and careful review comments. We thank Reviewer d1Yn for pointing out the strengths of our work, including
>
> * clear and understandable presentation
>
> * solid and comprehensive experiments
>
> * high performance
>
> * and novel from existing methods.
>
> We would like to answer the question as below:
>
> ---
>
> ___W2&Q2: I am curious about the performance of the proposed method when combined with other efficiency methods (e.g., quantization, pruning).___
>
> A2: We thank for this insightful question on the combination of our method and other kind of existing efficient methods.
>
> * __Pruning__
>
>   Since our method primarily focuses on reducing channel-wise complexity for ViTs, we decide to test our method combined with a spatial-wise token reduction method, ToME, rather than network pruning methods. The results of ToMe-RePa-ViTs with different token reduction numbers (r) are shown in the Table below. In general, our method combines with token reduction method well and yields more significant improvement in the trade-off between accuracy and efficiency.
>
>   __Notably, ToMe-RePa-ViT-Large/0.75 with reduction number 8 achieves more than 200% acceleration (374.7 imgs/s vs 107.7 imgs/s) with merely 0.8% accuracy drop.__
>
> |Backbone|r|Reparam|#MParam.|GMACs|Speed (imgs/s)|Accuracy|
> |-|-|-|-|-|-|-|
> |ToMe-RePa-DeiT-Small/0.75|0 (No ToMe)|×|22.1|4.6|1279.1|77.1%|
> ||0 (No ToMe)|√|13.2|2.9|1975.5|77.1%|
> ||1|√|13.2|2.7|1983.3|77.0%|
> ||2|√|13.2|2.6|2077.5|76.9%|
> ||3|√|13.2|2.4|2118.0|76.8%|
> ||4|√|13.2|2.2|2225.3|76.6%|
> ||5|√|13.2|2.1|2286.5|76.4%|
> ||6|√|13.2|2.1|2370.1|76.1%|
> ||7|√|13.2|2.0|2438.9|75.9%|
> ||8|√|13.2|1.9|2542.3|75.4%|
> |ToMe-RePa-DeiT-Base/0.75|0 (No ToMe)|×|86.6|17.6|393.8|81.8%|
> ||0 (No ToMe)|√|51.1|10.6|659.5|81.4%|
> ||1|√|51.1|9.6|660.6|81.5%|
> ||2|√|51.1|9.3|678.9|81.4%|
> ||3|√|51.1|9.0|697.4|81.4%|
> ||4|√|51.1|8.7|738.8|81.3%|
> ||5|√|51.1|8.4|746.9|81.2%|
> ||6|√|51.1|8.1|773.3|80.9%|
> ||7|√|51.1|7.8|812.0|80.6%|
> ||8|√|51.1|7.5|843.7|80.1%|
> |ToMe-RePa-ViT-Large/0.75|0 (No ToMe)|×|304.5|59.8|102.7|82.0%|
> ||0 (No ToMe)|√|178.4|34.9|207.2|82.0%|
> ||1|√|178.4|32.8|208.6|82.0%|
> ||2|√|178.4|30.7|220.9|81.9%|
> ||3|√|178.4|28.6|236.3|81.9%|
> ||4|√|178.4|26.5|257.1|81.8%|
> ||5|√|178.4|24.4|278.9|81.8%|
> ||6|√|178.4|22.3|304.9|81.6%|
> ||7|√|178.4|20.2|337.2|81.5%|
> ||8|√|178.4|18.1|374.7|81.2%|
>
> * __Quantization__
>
>   For simplicity, we employed TensorRT to perform FP32-to-FP16 quantization on our pretrained and reparameterized models. TensorRT provides a comprehensive ecosystem of tools designed to deliver high-performance deep learning inference via _post-training_ quantization. The evaluations before and after reparameterization of our RePaViT and RePaSwin models were conducted on the same hardware platform mentioned in the paper (NVIDIA A6000).
>
>   __Remakably, when quantized to FP16, our method presents more than 200% acceleration. And the acceleration even signifies with reparameterization, demonstrating the effectiveness of combining quantization with our method in real-world scenarios.__
>
>
> Model| Reparam | FP16 Imgs/s | FP16 Acc | FP32 Imgs/s | FP32 Acc
> -|-|-|-|-|-
> RePa-DeiT-Small/0.75|×|  11945.84 (+210.3%)  |  76.41%|  3849.89|  76.41%
> ||√|  18329.13 (+248.4%) |  76.40%|  5260.94|  76.41%
> RePa-DeiT-Base/0.75|    × |   3135.53 (+158.3%)  |  81.31%|  1213.87|  81.31%
> ||    √ |   5441.13 (+189.8%) |  81.32%|  1877.24|  81.32%
> RePa-ViT-Large/0.75 |    × |    935.75 (+172.6%) |  81.96%|   343.29|  81.96%
> ||    √ |   1650.12 (+210.8%) |  81.97%|   530.94|  81.95%
> RePa-Swin-Tiny/0.75 |    × |   7144.35 (+154.8%) |  78.43%|     2804.70 |  78.44%
> ||    √ |  13351.39 (+202.8%)  |  78.43%|   4408.99 |  78.43%
> RePa-Swin-Small/0.75 |    × |   4225.21 (+152.0%) |  81.56%|    1676.96 |  81.56%
> ||    √ |   7782.22 (+191.7%) |  81.56%|    2667.86 |  81.56%
> RePa-Swin-Base/0.75 |   × |    2670.20 (+152.8%) |  82.58%|    1056.28 |  82.59%
> ||  √  |4873.75 (+205.7%)  |  82.58%|    1594.54 |  82.58%
>
> __In conclusion, our method combines with quantization and pruning methods well, exhibiting more significant enhancement in performance-efficiency trade-off.__ And the integration of TensorRT indicates the potential of deploying on edge devices.
>
> ---
>
> ___W1&Q1: How does RePaViT perform in deployment scenarios on edge devices?___
>
> A1: We thank the Reviewer for this question, which is indeed interesting and essential for demonstrating the real-world practicality of our RePaViT.
>
> Unfortunately, due to strict equipment management policies in our organization and time limits, we were unable to successfully apply for access to an edge device during the rebuttal period. As a result, we could not provide real-world inference speed measurements on actual edge hardware. We would like to report detailed performance metrics upon deployment to the Jetson AGX Orin platform in the follow-up discussion phase.
>
> ---
>
> We hope our responses address your concerns and would sincerely appreciate your reconsideration of raising the score.

---

### Official Review · Reviewer_NyjX · 2025-03-14

**Overall Recommendation:** 3

**Summary:**

The paper introduces RePaViT, a method for accelerating Vision Transformers by applying structural reparameterization specifically to FFN layers. The key observation is that FFN layers significantly contribute to ViT inference latency, especially as the model scales. To address this, the authors propose a "channel idle" mechanism, maintaining a subset of channels inactive, forming a linear path that can be structurally reparameterized during inference. Experimental results demonstrate substantial speed-ups and even accuracy gains compared to the original ViTs. Notably, RePaViT achieves superior efficiency compared to state-of-the-art pruning and reparameterization methods, validating its practical value for real-world applications.

**Claims And Evidence:**

The primary claims—that FFN layers dominate ViT latency, and that structural reparameterization of these layers significantly improves computational efficiency—are strongly supported by comprehensive experiments.

**Essential References Not Discussed:**

The paper provides comprehensive references, and there were no obvious omissions of essential references directly relevant to the proposed method.

**Experimental Designs Or Analyses:**

The experimental design, including various model sizes and comparative baselines (vanilla ViT, pruning methods, reparameterization methods like SLAB), is robust. However, additional transparency about how exactly throughput measurements were standardized across different methods would enhance reproducibility.

**Methods And Evaluation Criteria:**

The proposed method—structural reparameterization applied directly to FFN layers via a channel idle mechanism—is clearly defined and justified by latency analyses. Evaluation criteria are appropriate and rigorously applied. Experiments with standard benchmarks like ImageNet, MSCOCO, and ADE20K provide credible validation.

**Other Comments Or Suggestions:**

See questions.

**Other Strengths And Weaknesses:**

Strengths:
1. The idea of structurally reparameterizing FFN layers is original and clearly presented.
2. Clear real-world application potential, especially given that many vision foundation models use ViT backbones.
3. Thorough and convincing experimentation demonstrating scalability, accuracy trade-offs, and latency improvements.

Weaknesses:
Batch norm instead of Layer norm may limit the training efficiency.

**Questions For Authors:**

1.  What motivated the default choice (0.75)? How sensitive is this choice in terms of generalization to other datasets/tasks beyond those tested?

2. Training Stability: Can you discuss any observed training instabilities or convergence issues with larger channel idle ratios or model scales, and how these might be mitigated in practice?

**Relation To Broader Scientific Literature:**

RePaViT effectively positions itself within current literature by contrasting clearly with:

1. Network pruning methods: emphasizing higher practical efficiency and hardware-friendliness.
2. Other reparameterization methods: highlighting methodological differences (vertical versus horizontal reparameterization) and targeting intrinsic ViT architectures rather than hybrid or CNN-augmented structures.

The paper situates its contribution well by emphasizing originality in applying structural reparameterization directly to FFN layers.

**Theoretical Claims:**

No proofs in the paper.

---

> ### Author Rebuttal · Authors · 2025-04-01
>
> We sincerely appreciate Reviewer NyjX for the recognition of our work, especially on
>
> * clear novelty
>
> * significant real-world application potential
>
> * and thorough and convincing experiments.
>
> We would like to answer and clarify the questions as below:
>
> ---
>
> ___Q1: What motivated the default choice (0.75)? How sensitive is this choice in terms of generalization to other datasets/tasks beyond those tested?___
>
> A1: We thank for this insightful question.
>
> Firstly, we would like to clarify that the channel idle ratio $\theta$ is a __static architectural hyperparameter__ designed to control the trade-off between model efficiency and performance. It is not meant to be universally optimal but rather user-configurable based on specific hardware constraints and application requirements. __More importantly, the choice of channel idle ratio is highly related to the model size__. As shown in Table 4, larger models (e.g., DeiT-Base) tolerate higher idle ratios with minimal accuracy drop. In particular, when the model size grows to DeiT-Base level and beyond, 0.75 idle ratio can result in high performance, which motivates our choice.
>
> Secondly, **as for the method-level generalization across tasks**, we would like to emphasize that RePaViT has been evaluated on four diverse tasks: (1) image classification on ImageNet-1K, (2) object detection on MS COCO, (3) semantic segmentation on ADE20K, and (4) zero-shot image classification using CLIP on LAION-400M. These results collectively demonstrate the method’s robustness and applicability across different computer vision problems.
>
> Thirdly, **as for the model-level generalization across datasets**, we have provided zero-shot classification results in Table 9, Appendix B, where RePaViT is applied to CLIP models with different idle ratios. Since CLIP is pretrained on the large-scale unlabelled LAION-400M dataset and directly evaluated on the ImageNet-1K validation set without finetuning, this setup reflects the model's generalizability on unseen data. Notably, RePa-CLIP-ViT-B/16 with an idle ratio of 0.5 can outperform the vanilla model by 0.5%.
>
> Finally, **on the sensitivity of the idle ratio**, Table 4 presents a detailed analysis. We observe that performance remains stable or improves slightly up to a certain point, and then degrades as the idle ratio increases. Specifically, when the idle ratio exceeds 0.75, a noticeable accuracy drop occurs. We hypothesize that excessive pruning may cause insufficiency of nonlinearity in the model, negatively affecting the model representation capacity.
>
> While our current study has provided extensive experiment results demonstrating the generalizability of our method and the sensitivity of idle ratio choice, we acknowledge that further evaluation could offer deeper insights into its robustness. However, conducting such experiments would require substantial additional time that can extend beyond the current rebuttal phase. We still welcome specific task/dataset suggestions from Reviewer NyjX and are happy to include additional results in the revised version.
>
> ---
>
> ___Q2: Training Stability: Can you discuss any observed training instabilities or convergence issues with larger channel idle ratios or model scales, and how these might be mitigated in practice?___
>
> A2: We appreciate this insightful question.
>
> Fortunately, we have not observed instability or convergence issues when training RePaViT with large idle ratios or larger models. As shown in Tables 1 and 4, and according to our training logs, RePaViTs converge smoothly across all configurations. We have also provided our source code and detailed instructions in the supplementary material for reproducing our results.
>
> However, __reparameterizing before training can cause instability__, particularly when BatchNorm is merged early. Table 5 shows that while such reparameterization reduces training time, it degrades performance and may destabilize training—especially for large models—since normalization layers are crucial for mitigating issues like internal covariate shift, ensuring consistent signal scales across layers, which is particularly important in larger models.
>
> ---
>
> ___W1: Batch norm instead of Layer norm may limit the training efficiency.___
>
> A3: We would like to clarify that BatchNorm is a dedicated architectural choice to facilitate a further reparameterization of normalization layers and shortcuts into the backbone. It enhances computational efficiency during inference, and we consider the increase in training time a worthwhile trade-off for the efficiency benefits achieved. Nonetheless, our method is still compatible with LayerNorm.
>
> ---
>
> We hope our responses address your questions and concerns, and would sincerely appreciate your consideration of raising the recommendation score.

---

### Decision · Program_Chairs · 2025-05-01

**Decision:**

Accept (poster)

**Comment:**

All reviewers support acceptance of this paper for the interesting and experimentally supported idea presented in this paper. So I m in favor of accepting this paper to ICML